# The Adult ADHD Self-Report Scale (ASRS): utility in college students with attention-deficit/hyperactivity disorder

Sarah Gray[1], Steven Woltering[1], Karizma Mawjee[1] and Rosemary Tannock[1,2]

[1] Applied Psychology and Human Development, Ontario Institute for Studies in Education, University of Toronto, Canada
[2] Neuroscience & Mental Health, The Hospital for Sick Children, Toronto, Canada

## ABSTRACT

**Background.** The number of students with Attention Deficit/Hyperactivity Disorder (ADHD) enrolled in colleges and universities has increased markedly over the past few decades, giving rise to questions about how best to document symptoms and impairment in the post-secondary setting. The aim of the present study was to investigate the utility and psychometric properties of a widely-used rating scale for adults with ADHD, the Adult ADHD Self-Report Scale (ASRS-V1.1), in a sample of post-secondary students with ADHD.

**Methods.** A total of 135 college students (mean age = 24, 42% males) with ADHD were recruited from Student Disability Services in post-secondary institutions. We compared informant responses on the ASRS administered via different modalities. First, students' self-report was ascertained using the ASRS Screener administered via telephone interview, in which they were asked to provide real-life examples of behavior for each of the six items. Next, students self-reported symptoms on the 18-item paper version of the ASRS Symptom Checklist administered about 1–2 weeks later, and a collateral report using an online version of the 18-item ASRS Symptom Checklist. Students also completed self-report measures of everyday cognitive failure (CFQ) and executive functioning (BDEFS).

**Results.** Results revealed moderate to good congruency between the 18-item ASRS-Self and ASRS-Collateral reports (correlation = .47), and between student self-report on the 6-item telephone-based and paper versions of the ASRS, with the paper version administered two weeks later (correlation = .66). The full ASRS self-report was related to impairment, such as in executive functioning (correlation = .63) and everyday cognitive failure (correlation = .74). Executive functioning was the only significant predictor of ASRS total scores.

**Discussion.** Current findings suggest that the ASRS provides an easy-to-use, reliable, and cost-effective approach for gathering information about current symptoms of ADHD in college and university students. Collateral reports were moderately related to self-reports, although we note the difficulty in obtaining informant reports for this population. Use of a telephone interview to elicit behavioral descriptions for each item may be useful in future research that is required to specifically test the utility of the ASRS in, for example, documenting and confirming current reports of impairment due to ADHD symptoms and its positive and negative predictive power for diagnosis.

Corresponding author
Rosemary Tannock,
rosemary.tannock@utoronto.ca

## INTRODUCTION

Attention-Deficit/Hyperactivity Disorder (ADHD) is a prevalent neurodevelopmental disorder that persists into adolescence and adulthood in about two-thirds of individuals (e.g., *Ebejer et al., 2012*; *Faraone, Biederman & Mick, 2006*), with an estimated prevalence in adults ranging from 1% to 6% (e.g., *Ebejer et al., 2012*; *Fayyad et al., 2007*; *Polanczyk & Rohde, 2007*; *Kessler et al., 2006*; *Simon et al., 2009*). In adulthood, ADHD is associated with substantial impairments in cognitive, academic, occupational, social, and economic functioning (e.g., *Biederman et al., 2008*; *de Graaf et al., 2008*; *Faraone et al., 2000*; *Kessler et al., 2005a*; *Kessler et al., 2005b*). These impairments pose unique challenges to a subgroup of adults with ADHD; namely, those in post-secondary educational settings. Attendance at college or university typically brings new challenges, including an abrupt decrease in external structure and support previously provided by parents, teachers, and others, combined with increased availability of immediate rewards and increased demands for behavioral self-regulation—areas in which individuals with ADHD are already vulnerable (*Fleming & McMahon, 2012*).

The past couple of decades have witnessed an increasing number of young adults with ADHD who gain entrance into the post-secondary education sector and register with college or university Disability Service Offices (DSOs) to request accommodations. Twenty five percent of post-secondary students who are receiving services at DSOs have an ADHD identification (*Wolf, 2001*). In the absence of epidemiological studies, the prevalence of ADHD in the post-secondary population is unknown, but estimates based primarily on self-reported diagnosis of ADHD or its symptoms range from 2% to 8%, depending on the criteria used (*DuPaul et al., 2009*). One critical issue for DSO staff is to be able to confirm that the student currently meets the DSM-IV or DSM-5 criteria for ADHD symptomatology, and that the student's report of current symptoms is robust. Documentation of impairment caused by the symptoms, is also essential for appropriate structuring and allocation of accommodations and technology.

Self-report measures are frequently used to confirm ADHD symptomatology and levels of impairment in college-aged students. Although research on assessment with this specific population is limited, the broader literature on diagnosis of ADHD in adults yields strong evidence that adults are reliable reporters of current ADHD symptoms (*Murphy & Schachar, 2000*) and that adults' self-ratings and informant ratings are highly correlated (e.g., *Downey et al., 1997*). However, findings are equivocal in terms of whether self-ratings or informant ratings are generally higher (e.g., *Katz, Petscher & Welles, 2009*; *Kooij et al., 2008*; *Zucker et al., 2002*). Accordingly, it has been suggested that multimodal assessment, including informant and self-report, should be used to gather more information about symptoms and impairments (*Alexander & Liljequist, 2013*). Moreover, clinical guidelines recommend that collateral report should be obtained and incorporated into the diagnostic

formulation of ADHD (*American Psychiatric Association, 2013*; *Canadian Attention Deficit Hyperactivity Disorder Resource Alliance, 2011*; *Kendall et al., 2008*).

The World Health Organization (WHO) Adult ADHD Self-Report Scale (ASRS) and its 6-item screener scale (*Kessler et al., 2005a*; *Kessler et al., 2005b*; *Kessler et al., 2007*) are standardized and well-validated tools for assessment of current ADHD symptoms in individuals aged 18 years and older (*Kessler et al., 2005a*; *Kessler et al., 2005b*; *Kessler et al., 2007*). We have chosen to use the ASRS scales in the current study, as they have not been investigated previously in studies of English speaking college and university students with ADHD (see the Supplemental Table). However, they are available in the public domain and hold potential for providing a cost-effective approach for confirming current symptoms of ADHD in college and university students, if found to yield reliable reports in this population.

Presence of symptoms and severity ratings gathered using scales such as the ASRS, are necessary but not sufficient for providing documentation at DSOs. Impairment in functioning at college due to ADHD symptoms must be confirmed and specified (*Canadian Attention Deficit Hyperactivity Disorder Resource Alliance, 2011*; *American Psychiatric Association, 2013*). Scales that measure impairments in executive functioning (EF) and highlight specific difficulties that impact functioning in everyday life can support this aspect of documentation. For example, difficulties with EF, attention, inhibition, reasoning, planning, and working memory, as well as cognitive failures that happen on a daily basis are very common in those with ADHD (*Willcutt et al., 2005*; *Barkley, Murphy & Fischer, 2010*). Therefore, in this study we have included measures of EF and everyday cognitive failure, in order to examine the relationship between these and ADHD symptoms as measured by the ASRS, in post-secondary students.

Our overall aim in the present study, was to investigate the utility and reliability of using the ASRS questionnaire supplemented by interview questions with an informant report to screen for ADHD symptoms in a sample of post-secondary students. To do so, we investigated the congruence between self-report and a collateral report by a significant other (e.g., parent, sibling, spouse, etc.), as well as the test-retest reliability of self-reported ADHD symptoms using different modalities across time, and congruency between the ASRS and self-report indices of executive functioning and cognitive failures in everyday life in college and university students presenting to DSOs.

## MATERIALS AND METHODS

### Participants

A total of 135 students with ADHD (age = 24, sd = 3.6; 57 males, 42%; 21% also registered with a learning disability [LD]) were recruited from University Disability Services via email lists and flyers. Participants would understand the need for measuring current symptoms in order to register as eligible for this intervention study, which was the focus of the larger-scale study. Inclusion criteria were; (1) current enrolment in a post-secondary program, (2) a previous diagnosis of ADHD, (3) registration with respective university or college Student Disability Services, which requires documented evidence of a previously

**Table 1 Sample descriptives reported separately by medication status.**

|  | Total ($n = 135$) | Non-medicated ($n = 56$) | Medicated ($n = 79$) |
|---|---|---|---|
| Age | 23.7 (3.6) | 23.5 (3.6) | 23.9 (3.5) |
| Female % | 78 (57.7%) | 31 (55.4%) | 47 (59.5%) |
| WASI | 111.6 (13.1) | 110.7 (13.7) | 112.5 (12.6) |
| K-10 | 37.0 (5.7) | 36.0 (5.5) | 37.9 (5.8) |
| **SA-45** |  |  |  |
| Anxiety | 62.1 (8.5) | 62.5 (8.7) | 61.7 (8.5) |
| Depression | 60.4 (7.2) | 61.3 (7.9) | 59.5 (6.3) |
| Obsessive compulsive | 70.2 (8.3) | 70.6 (8.3) | 69.7 (5.1) |
| Somatization | 55.7 (8.6) | 56.8 (9.3) | 54.7 (7.8) |
| Phobia | 62.8 (5.7) | 63.1 (6.1) | 62.6 (5.3) |
| Hostility | 58.3 (7.2) | 58.7 (7.8) | 57.8 (6.6) |
| Interpersonal sensitivity | 61.2 (7.5) | 60.7 (7.7) | 61.6 (7.2) |
| Paranoia | 56.6 (8.3) | 57.6 (8.9) | 55.6 (7.7) |
| Psychoticism | 61.3 (5.0) | 62.5 (5.8) | 60.3 (3.8) |
| Global severity index | 61.4 (8.1) | 62.0 (8.7) | 60.8 (7.6) |

**Notes.**

SA-45: T-scores, WASI: Standard Scores, K-10: raw scores.

confirmed diagnosis of ADHD (typically, but not invariably in elementary school), and (4) aged 19–35. Exclusion criteria were; (1) uncorrected sensory impairment, (2) major neurological dysfunction and psychosis, and (3) current use of sedating or mood altering medication.

Of this sample, 79 participants (59%) reported receiving medication for ADHD (age = 23.7, sd = 3.5; amongst whom 41% were male and 14% were also registered with DSO as having an LD) and 56 participants (41%) did not (age = 23.8, sd = 3.7; of whom 45% were male and 31% were registered with comorbid LD). As can be seen from the summary data in Table 1, students who were or were not receiving medication did not differ in age ($p = .78$), estimated IQ ($p = .28$), or current levels of psychological distress ($p = .08$), as determined by independent sample t-tests. Nor did the two groups differ in terms of sex distribution, Chi-square (1) $= .23$, $p = .63$, or in their reported scores on the psychopathy subscales (all $p$'s $> .06$). However, participants reported to be on medication were less likely to be registered with DSO as having comorbid LD, Chi-square (1) $= 4.90$, $p = .03$.

## Measures

*Adult ADHD Self-Report Scale-V1.1 Screener (ASRS-V1.1)*: The 6-item ASRS-V1.1 (*Kessler et al., 2005a*; *Kessler et al., 2005b*) was designed as a tool to help screen for ADHD in adults (aged 18 years and older). The 6 questions are consistent with the DSM-IV criteria and address the manifestation of ADHD in adults. The paper version requires 1–2 min to complete. Respondents are required to use a 5-item Likert scale to indicate the frequency of occurrence of symptoms (0 = never; 1 = rarely; 2 = sometimes; 3 = often; 5 = very often). According to convention, if the respondent has 4 or more responses marked in the dark-shaded boxes of the copyrighted paper-version of the Screener (or in Part-A

of the ASRS Symptom Checklist), then the current symptom profile of the individual is considered to be highly consistent with ADHD diagnosis in adults (*Adler et al., 2006*; *Kessler et al., 2007*). Using this scoring convention, previous studies (e.g., *Hines, King & Curry, 2012*) report high sensitivity (1.0) and moderate positive predictive power (0.52), suggesting that the ASRS would rarely miss ADHD in an adult who has ADHD. Moreover, the ASRS Screener has moderate specificity (.71) and high negative predictive power (1.0), indicating that this tool is quite successful in not identifying someone with ADHD when they do not have the disorder (*Hines, King & Curry, 2012*).

The data reported herein were derived from a telephone-based interview in which the interviewer (a trained psychology graduate student) administered the 6 questions of the ASRS-V1.1 Screener orally, with probes to elicit real-life examples of how the symptom typically manifests and its frequency (ASRS-TIPS; ASRS-Telephone Interview Probes for Symptoms). The ASRS-TIPS was always conducted before students or their significant-other completed the 18-item version of the ASRS-V1.1.

*Adult ADHD Self-Report Scale-V1.1 Symptoms Checklist (ASRS-V1.1)*: The 18-item ASRS v1.1 was designed to evaluate current manifestation of ADHD symptoms in people aged 18 years or older. This scale is based on the World Health Organization Composite International Diagnostic Interview ©2001, and the questions are consistent with DSM-IV criteria, but reworded to better reflect symptom manifestation in adults. This tool, which takes about 5 min to complete, has high internal consistency and concurrent validity (*Adler et al., 2006*). Part-A contains the same 6 items as in the Screener: Part-B contains 12 additional questions based on DSM-IV criteria. The original questionnaires are formatted with darkly shaded boxes in Part-A and Part-B: endorsements in the darkly shaded boxes signify more severe symptoms. For the purpose of this and the larger-scale study, we removed the darkly shaded boxes in the ASRS-V1.1 to minimize any possibility that the darker shaded areas may motivate symptom exaggeration. Henceforth, we refer to the boxes that would normally be darkly shaded, as 'criterion boxes'.

*ASRS-V1.1 for Other*: We modified the wording of the 18-item ASRS-V1.1 Symptoms Checklist to render it appropriate for completion by a student's significant other (i.e., parent, adult sibling, close relative or friend, or intimate partner). Also, all response boxes were white, meaning that there were no darkly shaded boxes. The tool was then uploaded onto a secure website (surveymonkey.com) for online completion by the significant other.

*Barkley Deficits in Executive Functioning Scale-Short Form (BDEFS)*: This Short Form, 2-item version of the BDEFS measures different constructs of everyday executive functioning in adults, including problem solving, organization, time management, self-regulation of emotions, self-restraint, and self-motivation. A summary score is produced from the sum of these subscales. A 4-point Likert-type scale is used to indicate frequency of occurrence (1 = never or rarely; 2 = sometimes; 3 = often; 4 = very often). Using clinical and college samples of adolescents and adults with ADHD, this scale has strong reliability and validity (*Barkley, 2014*; *Dehili, Prevatt & Coffman, 2013*).

*Cognitive Failures Questionnaire (CFQ)*: This self-report scale measures everyday failures of memory, motor functioning, and perception. Items are rated on a 4-point scale, quantifying the frequency of these mistakes (0 = never, 1 = very rarely, 2 = occasionally, 3 = quite often, 4 = very often). A total raw score is calculated from adding all 25 scores together. Studies have found that this scale predicts performance on laboratory measures of attention, and that it is a reliable scale (*Broadbent et al., 1982*; *Kanai et al., 2011*; *Forster & Lavie, 2007*).

*Symptom Assessment-45 (SA-45)*: The SA-45 was used as a brief assessment of a general psychiatric symptomatology (*Maruish, 1999*). Based on the well-validated longer version (SCL–90–R), this questionnaire consists of 45 items that ask participants the degree to which certain problems have bothered them in the past 7 days, using 5-point Likert-type scale ranging from not at all (1) to extremely (5). T-scores above 60 indicate elevated psychiatric symptoms.

*Wechsler Abbreviated Scale of Intelligence—Second Edition (WASI-II)*: The Vocabulary and Matrix Reasoning Subtests were used to estimate general intellectual ability (*Wechsler, 1999*).

*Kessler 10 Plus (K10+)*: The K10 is a self–report questionnaire which was used to obtain a global measure of nonspecific psychosocial distress. Questions probed the level of nervousness, agitation, psychological fatigue and depression in the past 30 days (*Kessler et al., 2002*). A higher score indicates more distress.

### Procedure

The data presented herein were derived from a larger-scale study investigating the behavioral and neural changes in college students with ADHD with a working memory training program (CIHR Grant #482246; Clinical Trials Registry #245899).

The study was approved by the Research Ethics Boards of the participating universities and colleges (protocol reference #23977). All participants provided informed written consent before starting the study. Participants were told explicitly that withdrawal from the study, failure to complete any components of the study protocol, and task performance would remain confidential and would not affect their DSO services or academic accommodations.

To confirm the robustness of students' self-report of current symptoms of ADHD, we used several procedures. In all cases, informants were instructed to complete their ratings based on when the student was in an unmedicated state (i.e., when medication effects had worn off or when they had forgotten or opted not to take medication over the weekend for example). First, we administered the six questions of the ASRS-TIPS orally by telephone (as part of the study intake procedure) and the student was asked to provide real-life examples for each of the six items, to ensure he/she understood the question and that the reported behavior was a reasonable example of an ADHD symptom. The interviewer prompted the student if needed for more details, clarification, or additional examples. The interviewer recorded the student's self-ratings on an ASRS form, along with the behavioral examples of symptoms provided by the student, but did not use this information to override the self-ratings.

Second, when each student came to the research lab for the first study assessment (T1), he or she was asked to complete both the 6-item Part-A (identical to the screener items) and the 12-item Part-B of the paper version of the ASRS (ASRS-v1.1), and to nominate and give us permission to contact a significant other who knew the student well enough to complete the ASRS (e.g., sibling, parent, or close friend). Third, the significant others completed a modified version of the 18-item 'ASRS-V1.1 for Others' using a secure, online software program (www.surveymonkey.com, n.d.). The significant others were made aware that the purpose of the study was solely for research aimed to investigate cognitive functioning and the effectiveness of a working memory intervention program.

At the baseline assessment (T1), participants also completed other questionnaires, including the Kessler 10 Plus (K10+: *Kessler et al., 2002*), an index of current levels of psychological distress, and the Symptom Assessment-45 (SA-45: *Maruish, 1999*) an index of psychopathology. After questionnaires were completed, neuropsychological and performance tests were administered, including the Wechsler Abbreviated Scale of Intelligence (WASI-II) as an estimate of intelligence. Several other measures were also administered but not reported in this study. For the purpose of this study, these measures are reported for characterisation of the sample, while the emphasis of analysis are related only to the ASRS.

For the first visit, participants were reimbursed for travel/parking costs ($25 CAD), with the knowledge that they would receive more substantial reimbursement ($150 CAD) for a second visit to be scheduled 2 to 3 weeks after the 5-week working memory training program, which was the focus of the larger study. Each visit required about 5 hrs to complete the full assessment, including the neural assessments, required for the larger-scale study.

### Analyses

IBM SPSS Statistics version 21 was used to conduct the statistical analyses. For the analyses comparing group differences in scores, mixed model repeated measures ANOVAs were run with Medication status as a between-subjects factor and Congruency as a within-subjects factor (e.g., congruency between Informant or Modality). Relationships between variables were examined using Pearson correlations. Effects of sex were also investigated, separately, in all analyses, as a between subjects factor. Partial eta-squared values (n2) were computed to ascertain effect size (ES). According to *Vacha-Haase & Thompson (2004)*, ES based on n2 = .01 corresponds to a small effect, n2 = .10 corresponds to a medium effect, and n2 = .25 represents a large effect.

## RESULTS

### ASRS Scores: basic descriptives

Most of the students completed both the interview- and the paper-versions of the ASRS (99.3% completed and one case had missing data for the ASRS paper-version, 97.8% completed and 3 missing for the ASRS-TIPS Interview). By contrast, only 44% ($n = 59$) of the students' nominated significant-other completed the on-line version of ASRS. For

**Table 2  Means and standard deviations for each of the ASRS versions reported by medication status and sex.**

|  | Total | Non-medicated | Medicated | Males | Females |
|---|---|---|---|---|---|
| ASRS-Self ($n = 134$) | 49.1 (9.2) | 48.5 (9.9) | 49.7 (8.5) | 46.1 (8.6) | 51.4 (9.1) |
| ASRS-Self Part-A ($n = 134$) | 17.6 (3.0) | 17.4 (3.1) | 17.8 (2.8) | 17.1 (3.2) | 18.0 (2.7) |
| ASRS-Interview ($n = 134$) | 17.5 (2.5) | 17.3 (2.5) | 17.7 (2.4) | 17.5 (2.7) | 17.5 (2.3) |
| ASRS-Other ($n = 59$) | 45.8 (10.9) | 43.6 (11.1) | 47.4 (10.6) | 45.0 (10.8) | 46.6 (11.1) |
| ASRS-Self[*] ($n = 59$) | 50.24 (9.8) | 49.3 (9.5) | 50.9 (10.1) | 48.6 (7.3) | 51.7 (11.5) |

Notes.

[*] Matched ASRS self-report subgroup.

the medication group, the response rate for significant-others was 59% ($n = 39$) and it was 41% ($n = 20$) for the non-medicated group. Thus, the analysis across reporter type will be conducted on this smaller subset of participants who have a collateral report. This subgroup did not differ from the rest in age ($p = .66$), estimates of IQ ($p = .43$), current levels of psychological distress ($p = .12$), ADHD symptom severity ($p = .22$) or any reported scores on the psychopathology subscales (all $p$'s $> .17$), as determined by independent sample t-tests. A Chi-square test also showed no group difference in sex distributions ($p > .28$).

Sex differences were found for the standard 18-item ASRS, as determined by independent samples t-tests, indicating that females reported a higher frequency of symptoms, $t(132) = 3.38$, $p = .001$. There were no main effects for sex (all $p$'s $> .10$) or medication status (all $p$'s $> .39$) on any other versions of the ASRS. Table 2 shows the means and standard deviations for the ASRS variables broken down for sex and medication status.

## Congruency across informant

The repeated measures ANOVA showed a significant effect of Informant, $F(1, 57) = 8.92$, $p = .004$, ES $= .14$, showing that the students' self-reported total score was significantly higher than that reported by their significant-other. However, as evident from the summary scores presented in Table 2, the mean scores on the 18-item ASRS Symptoms Checklist reported by both students and their significant-other far exceeded the threshold score of 29, indicating their scores were well above the 90th percentile (based on the distribution of scores in the general population). Medication status was not significant as a factor ($p = .74$). Similar analyses with Sex as a between-subject factor did not yield significant differences either ($p > .30$).

Most students received at least four responses marked in the 'criterion boxes' of the ASRS Other (98%) or Part-A of the ASRS Symptoms Checklist (98%). These findings suggest that for the majority of students, their current symptom profile, as reported by their significant-other or by themselves was consistent with an ADHD diagnosis in adults.

The partial correlation, controlling for IQ, between significant-other and self-report ASRS was significant, $r(59) = .46$, $p < .001$ (see Fig. 1). These data suggest that the reports of current symptoms by students and their significant others are moderately congruent. Moreover, as can be seen in the scatterplot (Fig. 1), the majority of the paired scores by

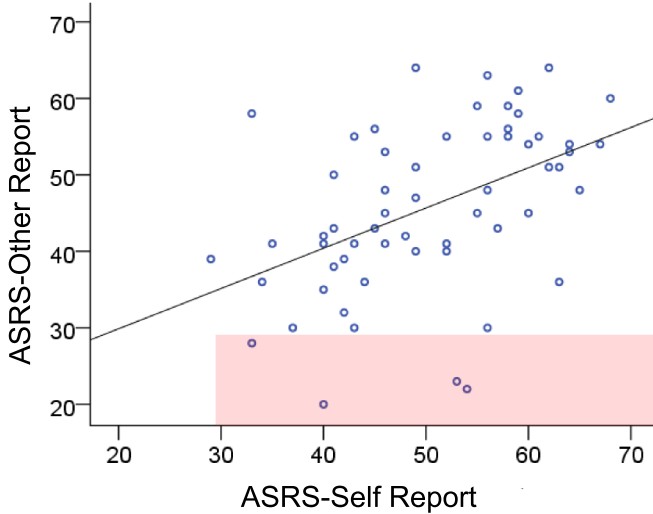

**Figure 1 ASRS self- and other-report.** Scatterplot showing paired ASRS scores for students and their significant-others. The pink shaded area indicates instances in which participants self-report above the 90th percentile (score 29) while significant others report it lower.

students and significant-others were above the 90th percentile (i.e., raw score on both axes were 29 and higher). Only 7% ($n = 4$) fell in the shaded area, which indicates the ASRS-Other scores that fall below the 90th percentile. Specifically, 97% of the ASRS-Other scores and 100% of the ASRS-Self-Report scores were at or above the 90th percentile.

## Congruency across ASRS Modality

The repeated measures ANOVA showed no significant effect of Modality, $F(1, 129) = .26$, $p = .61$, ES $= .002$, suggesting that there is no difference between the paper- and interview-versions of the self-reported ASRS Screener, despite the differences in modality and time between administration of the two versions. Medication status was not significant as a factor ($p = .34$). Similar analyses with Sex as a between subject factor did not find significant differences either ($p > .32$).

The correlation (controlling for IQ) between scores of self-reported symptoms across the ASRS-TIPS interview and Part-A of the ASRS paper version was significant, $r(131) = .66$, $p < .001$ (see Fig. 2), suggesting that the students' self-report of current symptoms was reasonably robust. Moreover, most respondents had at least four scores in the 'criterion boxes' on the ASRS-TIPS (100%) or Part-A of the ASRS Symptoms Checklist (98%), suggesting consistency between their current symptom profile, irrespective of modality (interview vs. pen-and-paper) and their documented diagnosis of ADHD as registered with DSOs.

## Congruency between the ASRS and measures of everyday cognitive failure and executive functioning

The total score from the 6-item ASRS screener (ASRS-TIPS) was moderately correlated with both the BDEFS EF summary score ($r(129) = .40$, $p < .001$) and the CFQ total score

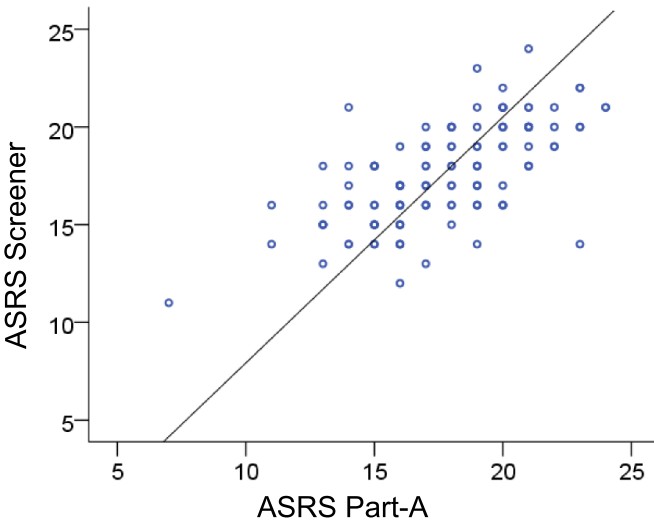

**Figure 2 ASRS interview and paper version.** Scatterplot showing the relation between the 6-item interview screener (ASRS-TIPS) and 6-item Part-A of the ASRS paper version.

($r$ (110) $= .46, p < .001$). The correlations between these scales were slightly stronger when using the ASRS 6-item paper version (CFQ: $r$ (112) $= .55, p < .001$; BDEFS: $r$ (131) $= .52$, $p < .001$). Of note, there was a stronger correlation between the full 18-item ASRS self-report total and both the BDEFS; $r$ (131) $= .62, p < .001$, and CFQ; $r$ (112) $= .74$, $p < .001$. This indicates that the full ADHD symptom scale covers more underlying constructs related to executive functioning and everyday cognitive failures as compared to the screener. The BDEFS measures specifically EF impairment, while the CFQ asks action orientated questions related to cognitive failure in everyday life; the two scales are strongly correlated: $r$ (111) $= .66, p < .001$. These two scales are not measuring ADHD symptomatology for diagnostic cut-off; rather, they measure functional impairment experienced by those with ADHD. The moderate-to-strong linear relationships indicate some portion of overlap between the ADHD symptom rating scale and EF and cognitive failure rating scales. However, as would be expected, all three scales measure different aspects of ADHD symptomatology. A positive correlation between the BDEFS and ASRS is consistent with the finding that ADHD groups score in the high range on the BDEFS EF summary score (96–98th percentile; Barkley, 2014). The BDEFS also best predicted the ASRS-Total score ($\beta = .53, p < .01$), over and above our measure of Psychopathology (SA-45; $\beta = 1.3$, n.s.) or Psychological Distress (K10; $\beta = .39$, n.s.) using a regression analysis.

The ASRS informant score was not significantly correlated with the CFQ self-report total score. There was a moderate positive correlation between the ASRS informant total score and the BDEFS self-report ($r$ (59) $= .36, p < .01$).

## DISCUSSION

To our knowledge, this is the first study to investigate the utility of a brief telephone-based interview to elicit college and university students' descriptions of their ADHD symptoms

in real life, used in conjunction with measures of EF impairment. It is also the first to investigate the test-retest reliability of English-speaking college students' self-reported current ADHD symptomatology, along with a collateral report.

The study yielded three major sets of findings: (1) Students' self-ratings of current ADHD symptoms were higher than scores reported by the significant other, but were moderately related to ratings by their significant-other; symptom ratings were all above clinical threshold for ADHD, regardless of informant or of the modality of administration; (2) Students' initial telephone-based ratings of symptom frequency were strongly related to their self-reported ratings on a paper-version of the questionnaire completed one to two weeks later; the majority of students met threshold criterion on the initial interview-based ASRS as well as on their second self-rating (on Part-A) one to two weeks later; and (3) Students' ratings of their ADHD symptoms on the 18-item ASRS were highly correlated with self-ratings of EF impairment and everyday cognitive failures. These correlations were not replicated between ASRS-other and self-report on the CFQ, however a moderate relationship was found between the ASRS-other and self-rated EF impairment. Scores of executive functioning were the only significant predictor of ASRS self-report.

On the ASRS, respondents report the frequency of occurrence of each of the symptoms (never, rarely, sometimes, often, very often). Symptom frequency is often associated with symptom severity, thus scores on the ASRS-V1.1 Symptom Checklist may also indicate the severity of ADHD. Our findings are consistent with previous research in college students that found similar patterns between self and informant ratings, that is, current self-ratings were higher than informant ratings (*Katz, Petscher & Welles, 2009*). However, in neither that study nor the present study, were these informant differences clinically meaningful.

The correlation between students' self-report and their significant-other-report revealed only modest congruence ($r = .47$), in this sample of college and university students. It is possible that the modest correlation was attributable in part to the smaller sample size: this analysis was based on only a subset of 59 participants with both self- and other-report. The low response rate by significant other might also be attributable to the fact that ratings were requested as part of a study rather than to validate symptoms for registration with DSO, so the motivation of the collateral informant may be lower. Future research is warranted on the feasibility of obtaining a collateral report for students in post-secondary education.

Good test-retest reliability between the ASRS-TIPS and the ASRS paper version indicate that this questionnaire is reliable when used over the telephone, supplemented with questions that elicit specific real-life examples of each symptom. There is growing concern that some students might feign or exaggerate symptoms of ADHD for personal gain, such as receiving academic accommodations, a waiver on student loan repayments, or to gain access to government-funded programs and services (*Diller, 2010*). Thus, use of the ASRS-TIPS to elicit examples of behavior may afford greater confidence in the validity of current symptoms reported by college students.

The positive linear relationship between the ASRS and measures of EF and cognitive failure in everyday life is consistent with the literature that emphasizes the impact of EF difficulties in adults with ADHD (*Willcutt et al., 2005*; *Barkley, Murphy & Fischer, 2010*).

Congruency between the ASRS and well validated and reliable scales of EF in college students add to this literature and indicate that both action-oriented (CFQ) and EF specific (BDEFS) measures tap into significant impairments related to ADHD symptoms in college students. The BDEFS was the only significant predictor of ADHD self-report scores. This indicates that impairment in EF, in this sample, is related to symptoms of ADHD and not as specifically to other psychopathology as measured by the SA-45 and K-10. This finding indicates that using these two measures together can increase knowledge about both ADHD symptoms and EF impairment related to these symptoms in college students.

To the best of our knowledge this is the first study to administer the ASRS-V1.1 Screener by telephone interview with probes for examples of how each symptom manifests in the student's daily life (ASRS-TIPS). Systematic and detailed analyses of the students' examples are in progress and will be reported elsewhere, but informal inspection of the behavioral examples indicated that the majority were excellent and valid examples of the specific symptoms. These data not only suggest that the students understood the question, but also afforded greater confidence in the robustness of their self-rating of the presence and frequency of occurrence of their current symptoms. Comparison of the students' self-ratings on frequency of occurrence on six ADHD symptoms during interview and on the paper-version of the ASRS revealed excellent stability and reliability of reporting across a one to two week interval, despite the differences in modality used to obtain the information (telephone interview without any visual support versus paper-version with written questions). This finding adds to the small body of literature indicating that adults can accurately self-report symptoms of ADHD. Administration of the ASRS 6-item Screener by telephone with probes for symptoms may increase confidence that symptom ratings are reflecting real life examples for the students. Moreover, the test re-test reliability of the ASRS was strong this college sample, suggesting that this tool may be useful for monitoring symptoms and severity across the semester.

## Limitations

It is essential to keep in mind the limitations of this study, while considering the findings. First, this sample of college and university students with ADHD may be biased: they were already registered with DSOs and were highly motivated to seek and undergo intervention, despite the heavy time commitment and effort required to complete the working memory training. Thus, the findings may not be generalizable to the population of students with ADHD in the post-secondary education sector. Results are also specific to this group of post-secondary students with higher cognitive and academic ability as compared to peers who do not go on to university or college. Second, we were unable to confirm whether the participants actually met the DSM-IV criteria for ADHD, but the fact that they were all registered with the college and university DSOs suggests that their documentation of an ADHD diagnosis and impairments was adequate. We were unable to confirm co-occurring disorders in this sample; we did not have access to information about confirmed diagnosis for other disorders, such as learning disabilities or psychiatric disorders. Third, we were unable to confirm who the significant-others were in many cases (i.e., parent,

sibling, partner, etc.) or that the students and their significant-others completed the 18-item ASRS-V1.1 questionnaire independently. However, that the students and their significant-others completed the questionnaire in different modalities (paper-version versus on-line electronic version) and at different time points (in the research lab versus on-line one to two weeks later), would have made it difficult for them to confer. Fourth, our comparisons of reported (self- versus significant-other) and modality (6-item screener items using interview or paper version) may be confounded by other factors, such as time or practice. It is also a limitation that we did not have informant reports for the BDEFS and CFQ. Finally, it is possible that financial incentive may have increased the student's motivation to exaggerate symptom severity to be included in the study. However, the incentive was not great ($25) given the length of baseline assessment plus preceding intake telephone call (about 5 h in total).

## Clinical and research implications

The ASRS is available in the public domain and provides a brief and cost-efficient tool that is readily administered by telephone, computer, or in paper format, to both the student and collateral informant. It is available in several languages and is recommended for clinical use internationally. In this sample, test-retest reliability across modality and time was strong, with moderate congruence between self- and other-report, indicating that more research is merited to further examine the use of the ASRS in gathering collateral information. The addition of probes for examples of each of the six ASRS Screener items in a telephone-interview version may afford the clinician greater confidence in the robustness of the student's self-report of current ADHD symptoms. Use in conjunction with self-report measures EF may provide valuable information about everyday impairment related to ADHD symptoms for college students.

## CONCLUSIONS

The 6-item ASRS Screener and 18-item ASRS Symptom Checklist are feasible, reliable and cost efficient approaches to use in the assessment and monitoring of ADHD symptoms in the college population. The use of probes to elicit examples of each symptom as manifest in daily life along with self-ratings, in combination with the inclusion of a collateral report and self-report of executive functioning and cognitive failure, may afford increased confidence of accurate symptom reporting and provide corroborating evidence for symptom severity and functional impairment.

### Funding

Funding for this project was provided by the Canadian Institute for Health Research (Tannock & Lewis; #482246) and the Canada Research Chairs Program (R Tannock). The funders had no role in study design, data collection and analysis, decision to publish, or preparation of the manuscript.

### Grant Disclosures

The following grant information was disclosed by the authors:

Canadian Institute for Health Research: #482246.

Canada Research Chairs Program.

### Competing Interests

Dr Tannock is on the advisory board for Eli Lilly, a consultant for Purdue Pharma Canada and Shire, and in 2010 participated in an ADHD meeting sponsored by Janssen-Cilag. Other authors do not declare any competing interest.

### Author Contributions

- Sarah Gray conceived and designed the experiments, wrote the paper, prepared figures and/or tables, reviewed drafts of the paper.
- Steven Woltering conceived and designed the experiments, performed the experiments, analyzed the data, prepared figures and/or tables, reviewed drafts of the paper.
- Karizma Mawjee conceived and designed the experiments, performed the experiments, reviewed drafts of the paper, data collection.
- Rosemary Tannock conceived and designed the experiments, wrote the paper, reviewed drafts of the paper.

### Human Ethics

The following information was supplied relating to ethical approvals (i.e., approving body and any reference numbers):

The present study was approved by the following Research Ethics Boards: Centennial College, REB Application #135 (The Engage Study); Humber College, REB protocol #0193; Ryerson University, REB protocol: 2012-227; University of Toronto & York University: Protocol #23977. All participants provided informed written consent prior to the start of the study.

### Supplemental Information

Supplemental information for this article can be found online at http://dx.doi.org/10.7717/peerj.324.

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
