# Peer review of "The Adult ADHD Self-Report Scale (ASRS): utility in college students with attention-deficit/hyperactivity disorder"

_PeerJ, doi:10.7717/peerj.324_

## Round 0.1 · original submission · Major Revisions

Dear Authors,Please do the relevant (major revisions) as required by the two peer reviewers.Do sent them as soon as possible so as the manuscript can be re-reviewed by same people.

Reviewer 1 ·

Basic reporting

This report has sufficient introduction and literature background of the study. However, in reading this articles towards the end, the reviewer feels that the author may have tendency to suggest the reviewer to conform with the study (to agree Adult SRS is a good measurement in assessing ADHD in college students) rather than helping reviewer to understand the relationship the results with other measurement tools used in the study.

Experimental design

The author has a variety ways in getting information such in telephone interview, self-reporting and getting information from siginificant other. However, the author may need to carefully control the other variables such as range of IQ and range of psychiatric of symptomatology. This article could be better clarified as Adult ADHD Self Report Scale may have better outcomes in High Average college students (as author reported all the college students in this study only grouped in high average range, and not represent other IQ range such as average group or superior group).

Validity of the findings

The range of depression, obsessive compulsive and interpersonal sensitivity in this study has fallen into moderate range but it seems from analysis that it does not affect the overall data in Adult ADHD Self Report Scale which may rise a question to what extent the psychiatric symptomatology may interact with ADHD symptoms as we might predict some patients with ADHD has comorbidity with other psychiatric symptoms and rarely we could find such a large number of "pure" ADHD cases.

Additional comments

The author has done a great research in coming up with these data. However, the author may need to improve in discussing of all other findings that may have interact with the main objective of the study.

·

Basic reporting

This study addressess "the robustness of student's self-reports of current ADHD-symptoms", related to question to what extend "the student's report of current symptoms is robust and that the reported symptomatology is not exaggerated or feigned".
135 students (mean age 24; 42% males) were included; 59 (=44%) were included in the analyses.

My main concern with this study is that I doubt whether the central research question can be addressed with the current study design.

In order to know more about "exaggerating symptoms" in a given disorder, it should be clear that the disorder is present or is not present.
As there are no details on how the diagnosis of adult ADHD is obtained, this remains unclear.

It seems to me the study addresses 1)test-retest reliability and 2) the external validity of the ASRS.
In my opinion the central research question is not addressed.

So either the paper should be rewritten, in order to have the main research question related to test-retest rliability and external validity of the ASRS in this population, or another design is necessary.

As adult ADHD is a chronic condition that does not dissapear, it seems to me not logic why students with a confirmed diagnosis of ADHD would exaggerate or feign their symptoms? So the underlying research question is whether or not the ADHD diagnosis is true or false. And this cannot be answered with the given instruments/design.

So in my view the only way the information can be used is rewrite the manuscript and focus on test-retest reliability and external validity of the ASRS in this population.

Experimental design

1) It is unclear why the information on all of the included instruments that are used in the more extended study, are reported here. It is confusing;

2) Line 247 speaks of the "5-week working memory training program". Where does this come from, and why is this information given?

3) I suggest the authors to elaborate more on the consequences of the high drop-out rate. It seems to me that it might be that the 56% of cases that did not have a responding significant other might be related to ADHD symptom severity, or to "exaggerating symptoms"?
I have too little statistical knowledge to think of the consequenses of this drop-out rate, related to the presented results. So I suggest the Editor to have a statistical expert look at the methods and results.

4) What did the participants and their significant others know about the purpose of this study? For knowledge on the purpose might have influenced the results?

5) To learn more about the reliability of reporting on ADHD-symptoms it seems weird that ADHD-medication should not be taken into account. If students do use ADHD medication, and they do so for a considerable period, they are asked to report on the ADHD symptoms of a long time ago. Hence, the conclusion that the ASRS can be used for monitoring ADHD symptoms cannot be drawn from this study, for it aks in a considerable part of the group, for ADHD-symptomatology in the past.

Validity of the findings

No additional comments: see the above mentioned issues.

Additional comments

No additional comments.

Reviewer 3 ·

Basic reporting

This is an excellent paper that is both clear and concise. I commend the authors for their work in this area and believe the paper should be published in PeerJ.

Experimental design

No comments -- appropriate experimental design.

Validity of the findings

The findings are valid and placed in appropriate context.

---

## Round 0.2 · accepted · Accept

Thank you for submitting the revised manuscript which has been reviewed favorably and accepted for publication.

Reviewer 1 ·

Basic reporting

The article has sufficient introduction and background.

Experimental design

Methods has been described with sufficient information.

Validity of the findings

The conclusion has been stated appropriately.